# Adolescent undernutrition in South Asia: a scoping review protocol

Sara Estecha Querol  ,[1,2] Lena Al-Khudairy,[3] Romaina Iqbal,[4]
Samantha Johnson,[5] Paramjit Gill[1,2]

[1]Academic Unit of Primary Care, University of Warwick Warwick Medical School, Coventry, UK
[2]NIHR Global Health Research Unit in Improving Health in Slums, University of Warwick Warwick Medical School, Coventry, UK
[3]Division of Health Sciences, University of Warwick Warwick Medical School, Coventry, UK
[4]Department of Community Health Sciences and Medicine, Aga Khan University, Karachi, Pakistan
[5]University of Warwick Warwick Medical School, Coventry, UK

**Correspondence to**
Sara Estecha Querol;
Sara.Estecha-Querol@warwick.ac.uk

## ABSTRACT

**Introduction** The aim of the protocol is to present the methodology of a scoping review that aims to synthesise up-to-date evidence on adolescent undernutrition in South Asia.

**Methods and analysis** The proposed scoping review will be guided by Arksey and O'Malley's framework and the Joanna Briggs Institute Reviewers' Manual. The scoping review question, eligibility criteria and search strategy will be based on the Population, Concept and Context strategy. We will conduct the search in electronic bibliographic databases (Medline (OVID), Embase, Cochrane Library, Web of Science, CINAHL, PsycInfo, Scopus) as well as various grey literature sources in order to synthesise and present the findings with descriptive statistics and a narrative description of both quantitative and qualitative evidence.

**Ethics and dissemination** This study protocol does not require ethical approval. This protocol will accurately describe the proposed scoping review that will map the evidence on adolescent undernutrition in South Asia. The proposed review aims to gather published and unpublished literature to inform policy and healthcare organisations as well as identify future research priorities in South Asia.

## Strengths and limitations of this study

► This protocol will ensure transparency in methodology and reduce the likelihood of reviewing bias.
► This protocol will clearly describe the proposed methods to collect and synthesise evidence on adolescent undernutrition in South Asia.
► The proposed search strategy will be conducted in seven electronic databases.
► Grey literature sources will be also included in the proposed scoping review, such as government reports and organisation websites.
► There will be no quality assessment of the included studies.

## INTRODUCTION

The most recent Global Nutrition Report highlights the global burden of malnutrition and that it remains unacceptably high affecting every country in the world.[1] The World Health Assembly formulated six global nutrition targets aiming to tackle the burden of malnutrition worldwide by 2025.[2] These targets plan to decrease stunting and wasting in children, reduce anaemia in women, decrease the rates of low birth weight, increase the rates of exclusive breastfeeding in the first 6 months and ensure that there is no increase in childhood overweight. Malnutrition refers to deficiencies, excesses or imbalances in a person's intake of energy and/or nutrients.[3] Focusing on early stages in life, the term malnutrition covers several conditions affecting child growth patterns such as stunting, wasting, underweight, micronutrient deficiencies and overweight/obesity. Stunting, or low height for age, is caused among other factors, by long-term insufficient nutrient intake, frequent infections and

diseases. Wasting, or low weight for height, also indicates malnutrition prevalence among children under 5 years of age. While wasting is the result of acute significant food shortage and/or disease, stunting represents chronic malnutrition and the effects are largely irreversible. Underweight, or low weight for age, includes children under 5 with low weight for height (wasting) and low height for age (stunting) and considered a proxy indicator for undernutrition if data on wasting is not available. Micronutrient deficiencies or insufficiencies, also called 'hidden hunger', indicate a lack of important vitamins and minerals. Stunting, wasting, underweight and micronutrient deficiencies constitute the term 'undernutrition'.[3] Overweight/obesity is another form of malnutrition that generally involves the development of noncommunicable diseases. Overweight/obesity are attributed to the term 'overnutrition'.[3] However, grown reference data and indicators are different between children and adolescents. The WHO recommends the use of BMI for age (thinness and overweight) and height for age (stunting) to explore adolescent growth patterns.[4 5] Stunting is defined as height-for-age<2 SDs below the WHO Child Growth Reference median. Overweight is defined as BMI-for-age>1 SD above the WHO Growth Reference median, and obesity is

BMJ

defined as >2 SDs above the WHO Growth Reference median. Thinness is defined as BMI-for-age<2 SDs below the WHO Growth Reference median.[5] The term thinness and underweight have been used interchangeably when making reference to adolescents' low BMI-for-age.[6 7]

The *State of food security and nutrition in the world* reviewed the global prevalence of thinness among children aged 5–9 years and adolescents aged 10–19 years.[8] Thinness (BMI for age below 2 SD) in these age groups was associated with a higher risk of infectious diseases, delayed maturation, reduced muscular strength, work capacity and bone density later in life. In addition, thinness in adolescent girls was associated with adverse pregnancy outcomes and intra-uterine growth retardation. Globally, 10% of children aged 5–19 have a BMI-for-age below 2 SD. There are significant differences in the prevalence of thinness among children aged 5–19 years by region of the world. For instance, the prevalence is high at >15% in South Asian countries such as India, Afghanistan, Bangladesh, Bhutan, Nepal, Pakistan and Sri Lanka in comparison to <2% in Latin America and the Caribbean, Northern America, Europe and Oceania. The global prevalence of thinness remained fairly steady over the past decade.[8] The prevalence of stunting (height for age below 2 SD), thinness (BMI for age below 2 SD) and overweight (BMI for age above 1 SD) were globally estimated using data from the Global School-Based Student Health and Health Behavior in School-Aged Children surveys conducted in 57 low and middle income countries between 2003 and 2013, involving 129 276 adolescents aged 12–15 years.[9] Globally, the prevalence of stunting was 10.2%, thinness was 5.5% and overweight 21.4%. Particularly in Pakistan, the prevalence of stunting was 7.1%, thinness was 11.2% and overweight 6.4% in 2009. In India in 2007, the prevalence of stunting was 14.6%, thinness was 15.9% and overweight 11.1%. In Sri Lanka in 2008, the prevalence of stunting was 25.6%, thinness was 31.5% and overweight 4.8%.

There are around of 1.8 billion adolescents worldwide and the majority are clustered in low-income and middle-income countries.[6] However, there is a gap in adolescent health data.[10] While the large global health and nutrition surveys mainly collected household data from adolescent girls aged ≥15 years; the 10–14-year-old age group, unmarried adolescents, unschooled adolescents and male adolescents have not been included in global health surveys.[10] Christian and Smith[6] highlighted the public health need to investigate adolescent nutrition deficiencies, growth and development.

Child and adolescent undernutrition in South Asia was previously reviewed[11–14] (see a summary of reviews in table 1). Some literature reviews[11 13 14] were focused on children, while Mak's systematic review[12] provided evidence on both children and adolescents. However, Mak's systematic review exclusively focused on studies examining the prevalence of underweight status in the last two decades. This scoping review protocol will include a larger pool of evidence by proposing a broader search strategy including micronutrient deficiencies as well as including grey literature without restrictions on date of publication.

**Table 1** Reviews on child and adolescent undernutrition in South Asia

| | Akhtar[11] | Mak[12] | Pasricha[13] | Khan[14] | The proposed scoping review |
|---|---|---|---|---|---|
| Type of study | Literature review | Systematic review | Literature review | Literature review | Scoping review |
| Date of publication the studies included | Restrictions on date of publication not specified | 1990–2010 | Restrictions on date of publication not specified | Restrictions on date of publication not specified | Up to 2019 |
| Context of the population | South Asia (India, Pakistan, Bangladesh, Sri Lanka and Nepal) | East Asia (China, Hong Kong, Japan, South Korea, and Taiwan), South Asia (Bangladesh, India, Indonesia, Malaysia, Nepal, Pakistan, Singapore, Thailand, and Vietnam), and West Asia (Bahrain and Iran). | South and South-East Asia (no countries are listed) | South Asia (no countries are listed) | South Asia (Afghanistan, Bangladesh, Bhutan, India, Maldives, Nepal, Pakistan, and Sri Lanka) |
| Age of population | Children under 5 years and school children (up to 12 years) | Children and adolescents | Children under 5 years | Children under 5 years | Adolescents 10–19 years |
| Concept of malnutrition | Undernutrition | Underweight | Undernutrition | Undernutrition | Undernutrition |

## REVIEW AIMS

The aim of this protocol is to describe the methodology of an up-to-date scoping review of the literature on adolescent undernutrition in South Asia and identifying gaps in knowledge. The proposed scoping review work could potentially be valuable to inform policy makers and healthcare organisations delivering adolescent health and nutrition actions in South Asia. It will constitute the first step in a larger research project aimed at improving adolescent health in Pakistan; these findings will inform the next quantitative and qualitative phases.

## METHODS AND ANALYSIS

All study types will be included to ensure the inclusion of the majority of relevant literature.[15] The design of the proposed scoping review methodology was informed by Arksey and O'Malley's framework[15] and The Joanna Briggs Institute Reviewers' Manual.[16] Arksey and O'Malley's methodological framework includes five stages to conduct a scoping review.

### Stage 1: Identifying the research question

We will use the Population, Concept and Context (PCC) strategy to define the title, scoping review objective, scoping review question, and inclusion criteria.[16] Therefore, our main research question will address: adolescent undernutrition in South Asia, what do we know from the existing literature?

### Stage 2: Identifying relevant studies

At this stage, the reviewing team (SEQ, SJ, RI, LA-K, PG) discussed the key words constituting the search strategy and the criteria for inclusion and exclusion of the studies in accordance with the PCC strategy.

#### Databases and search strategy

An experienced research librarian (SJ) and one reviewer (SEQ) developed the search strategy by testing the keywords, MESH terms and databases to search. Finally, the reviewing team discussed and determined the final search strategy. The search will be conducted using the following electronic bibliographic databases and grey literature sources: Medline (OVID), Embase, Cochrane, Web of Science, CINAHL, PsycInfo, Scopus, the WHO Library Information System (WHOLIS), eLENA e-Library of Evidence for Nutrition Actions, and Opengrey. The websites of agencies, academic institutions and technical bodies will be also be searched: World Health Organisation (WHO), United Nations International Children's Emergency Fund (UNICEF), Demographic and Health Surveys (DHS), Program, Planning and Development Department AJ&K, Global Health Data Exchange (GHDx), World Food Program (WFP) and World Bank eLibrary. In addition, the topic field expert (RI) will be consulted to identify additional relevant grey literature sources. One reviewer (SEQ) will search additional resources by hand searching the references of included studies to identify further relevant evidence. This search strategy includes South Asia AND adolescents AND undernutrition. Related terms to undernutrition were also considered in the search strategy such as stunting, thinness, underweight and micronutrient deficiency. The search strategy was tailored to the specific requirements of each database. No restrictions on language or date of publication will be made. See full search strategy in online supplementary appendix 1.

#### Eligibility criteria

Despite the broadness of the proposed scoping review research question, the eligibility criteria should be clearly defined in order to thoroughly guide the reviewers' decisions. The inclusion criteria listed below are based on the PCC strategy:

► Population: adolescents mean age 10–19 years old.[17]
► Context: South Asia. The World Bank limits the South Asia region to Afghanistan, Bangladesh, Bhutan, India, Maldives, Nepal, Pakistan and Sri Lanka.[18]
► Concept: in this proposed scoping review only undernutrition indicators for adolescents will be used (stunting, thinness or underweight, and micronutrient deficiencies[3]). Adolescent undernutrition will be assessed quantitatively using the following WHO indicators: (1) thinness or underweight (low BMI-for-age); (2) stunting (low height-for-age) and (3) micronutrient deficiencies. All studies will be included regardless of the growth references or cut-off values for micronutrients deficiency followed.
► Adolescent undernutrition in South Asia will also be assessed qualitatively, therefore qualitative studies exploring perspectives, experiences or opinions around adolescent undernutrition will be included.
► Type of studies: quantitative and qualitative studies as well as grey literature, for example, primary research studies, reviews, government reports and guidelines.

Studies will be excluded if they include:
► Only overnutrition indicators (obesity and overweight).
► Pregnant or breastfeeding adolescents.
► Adolescent athletes.
► Adolescents with long term conditions such as diabetes, tuberculosis or HIV.
► Hospitalised adolescents.
► Intervention studies targeting treatment of a specific illness or condition such as diarrhoea.

### Stage 3: Study selection

The screening process of this proposed scoping review will comprise two phases. First, titles and abstract will be reviewed by two independent reviewers following a broad inclusion criteria, that is, studies looking at adolescent undernutrition in South Asia. Papers identified by either or both reviewers will be included in the next phase, which is the full-text screening. The same reviewers will screen full-text studies using the eligibility criteria mentioned above. Disagreements will be resolved by either discussion

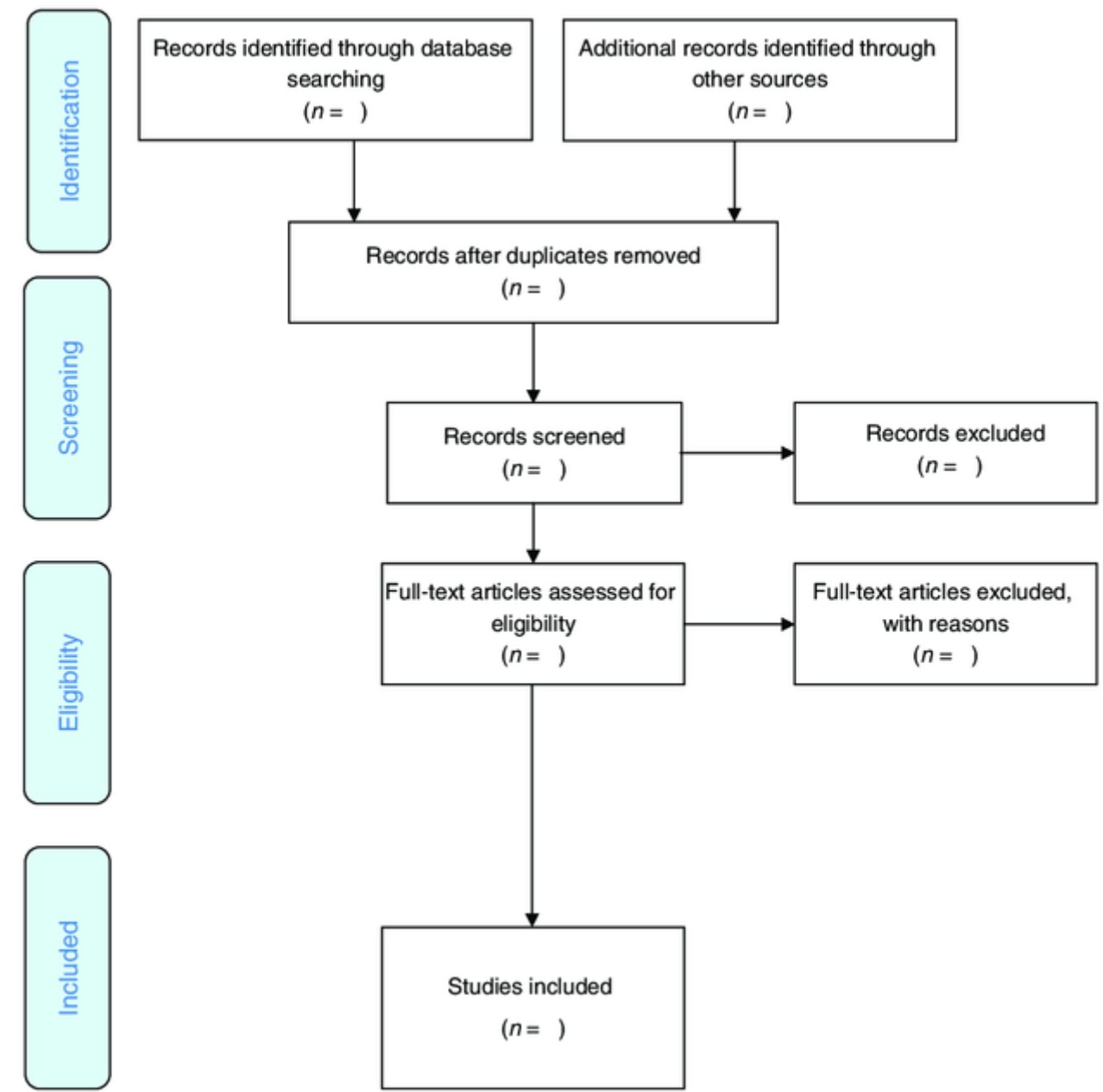

**Figure 1** Preferred Reporting Items for Systematic Reviews and Meta-Analyses flow diagram for scoping review process. Retrieved from Peters *et al.*[16]

or referral to a third reviewer. Preferred Reporting Items for Systematic Reviews and Meta-Analyses flow diagram (see figure 1) will be presented to reflect the search process.

### Stage 4: Charting the data

Key characteristics to be extracted from the included studies will be extracted using a predefined data-extraction sheet. At this stage, one reviewer (SEQ) will conduct data extraction consisting of a summary table recording the key information of the selected studies (see box 1).

### Stage 5: Collating, summarising and reporting the results

The results of the review will be reported as a map of the data extracted in a tabular form showing descriptive statistics. The tables may display the results as distribution of studies per year, geographical area where the study was conducted, location (community, hospital or school based), target population, outcomes and methods. In addition, a narrative description of both quantitative and qualitative results will accompany the descriptive presentation of the results by grouping the data into meaningful summaries for better exposition of the findings. For the qualitative studies, thematic analysis will be used to analyse the findings if data permits. Data will be exported into Nvivo and an inductive coding approach will be followed.

### Patient and public involvement

No patient involved.

## Box 1   Key information to be extracted

**Study details**
► Author(s)/organisation.
► Year of publication.
► Country.
► Province and city (where the study was published or conducted).
► Study design.
► Aims.
► Sample size.

**Data relevant to the PCC term**
► Population: target population (age, sex).
► Context: country.
► Concept: outcome quantitative measures (stunting, thinness or underweight, and micronutrient deficiencies). Also, note the growth references or the cut-off values for micronutrients deficiency.
► Concept: qualitative outcomes (experiences, opinions and perspectives).
► Summary of findings.
► Recommendations of the author(s)

## ETHICS AND DISSEMINATION

Since a scoping review involves a methodical integration and presentation of available resources, this study does not require ethics approval.

This protocol will provide a high quality overview of the broad literature available on the adolescent undernutrition in South Asia. Thus, the proposed scoping review will methodologically map the evidence on this topic to identify research priorities in South Asia. The proposed review will also facilitate the identification of uncovered disparities among adolescents living in South Asia. The potential gaps could help policy makers and healthcare organisations to design new research questions as well as interventions to improve adolescent malnutrition in South Asia.

**Contributors**  SEQ conceived the idea, developed the methods and wrote the first draft of the manuscript. PG, LAK and RI supervised the protocol, contributed to methods, and supported the drafting and editing the manuscript. SJ contributed meaningfully to design the search strategy (appendix 1). All authors revised and approved the final manuscript.

**Funding**  This work was supported by NIHR grant number 16/136/87. The research was commissioned by the National Institute of Health Research using Official Development Assistance (ODA) funding. The views expressed are those of the author(s) and not necessarily those of the NHS, the NIHR or the Department of Health and Social Care.

**Competing interests**  None declared.

**Patient consent for publication**  Not required.

**Provenance and peer review**  Not commissioned; externally peer reviewed.

**ORCID iD**
Sara Estecha Querol http://orcid.org/0000-0003-2018-2676

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
