## [Reviewer comments · BMJ Open]

ARTICLE DETAILS

TITLE (PROVISIONAL)	Adolescent undernutrition in South Asia: a scoping review protocol
AUTHORS	Estecha Querol, Sara; Al-Khudairy, Lena; Iqbal, Romaina; Johnson, Samantha; Gill, Paramjit

VERSION 1 – REVIEW

REVIEWER	Professor Premananda Bharati Biological Anthropology Unit, Indian Statistical Institute, Kolkata, India
REVIEW RETURNED	26-Jun-2019

GENERAL COMMENTS	This may be a general protocol for any systematic review using PRISMA guidelines. I think the outcome may consider for publication rather than to publish the study protocol.
---

REVIEWER	Dr Elizabeth M. Joseph-Shehu Department of Nursing Science, Faculty of Health Sciences, National Open University of Nigeria, Abuja, Nigeria
REVIEW RETURNED	28-Nov-2019

GENERAL COMMENTS	The protocol is well written and the study can be repeated but the authors need to be consistent with the concept (undernutrition) under study - objective is not well stated in the abstract. it should be stated as this paper presents a scoping review protocol on adolescent malnutrition in South Asia to identify..... - under review aim, line 48adolescent malnutrition.... while stage 1, research question..... addressed adolescent undernutrition.... be consistent with the one you are interested in i think the concept here is undernutrition and not malnutrition.
---

REVIEWER	Hanan Khalil La Trobe University
REVIEW RETURNED	03-Dec-2019

GENERAL COMMENTS	Thank you for the opportunity to review the manuscript. I have the following minor comments for the authors to consider. 1. Please include two proposed tables for data extraction; one for study details and the other for data relevant to the PCC term 2. Please remove table 1 from the introduction, it is not needed. 3. The aim of the protocol is to describe the methodology of the review that is aiming at mapping the evidence addressing adolescent malnutrition in South Asia, please update your aims. 4. Please consider using one scoping review methodology and adhere to it rather than making references to 4 or 5 of them as they are not the same.
--

5. Please consider including examples of the proposed tables for data extraction.

VERSION 1 – AUTHOR RESPONSE

Reviewer: 1

Reviewer Name: Professor Premananda Bharati
Institution and Country: Biological Anthropology Unit, Indian Statistical Institute, Kolkata, India
Please state any competing interests or state 'None declared': None Declared

Please leave your comments for the authors below
This may be a general protocol for any systematic review using PRISMA guidelines. I think the outcome may consider for publication rather than to publish the study protocol.

Thank you Prof Bharati for reviewing this scoping review protocol. The research team believes that it is important to publish the protocol and not only the scoping review for a list of reasons. Firstly, peer-review process of the protocol gives us valuable feedback and constructive feedback to improve our research. Secondly, publishing the protocol will ensure transparency in methodology and reduce the likelihood of reviewing bias. Thirdly, publishing this protocol will provide detailed methodology because the anticipated published review will require a specified word limit. Finally, scoping review protocols cannot be registered in PROSPERO, so publishing them in a peer-reviewed journal avoids duplication of similar work.

Reviewer: 2

Reviewer Name: Dr Elizabeth M. Joseph-Shehu
Institution and Country: Department of Nursing Science, Faculty of Health Sciences, National Open University of Nigeria, Abuja, Nigeria
Please state any competing interests or state 'None declared': None declared

Please leave your comments for the authors below
The protocol is well written and the study can be repeated but the authors need to be consistent with the concept (undernutrition) under study
- objective is not well stated in the abstract. it should be stated as this paper presents a scoping review protocol on adolescent malnutrition in South Asia to identify.....
- under review aim, line 48adolescent malnutrition.... while stage 1, research question..... addressed adolescent undernutrition.... be consistent with the one you are interested in
i think the concept here is undernutrition and not malnutrition.

Thank you Dr Joseph-Shehu for reviewing this protocol and for pointing out the consistency of the concepts. As you mentioned, the concept is undernutrition, so I have made the necessary changes (highlighted in yellow). I have also clarified the objective in the abstract (highlighted in yellow).

Reviewer: 3

Reviewer Name: Hanan Khalil
Institution and Country: La Trobe University
Please state any competing interests or state 'None declared': None Declared

Please leave your comments for the authors below
Thank you for the opportunity to review the manuscript. I have the following minor comments for the authors to consider.

Thank you Dr. Khalil for taking the time to review this protocol and for your useful comments that helped improve the overall reporting and content of the protocol. Please find our response to your

comments:

1. Please include two proposed tables for data extraction; one for study details and the other for data relevant to the PCC term

Thank you for this, we have now included this table as Table 2 (at the stabe 4).

2. Please remove table 1 from the introduction, it is not needed.

Thank you for your suggestion, we included this table to provide a narrative visual summary on the nature of published literature so far. This table supports that rational for the conduct of this work that is not presented in detail within the text due to word limits.

3. The aim of the protocol is to describe the methodology of the review that is aiming at mapping the evidence addressing adolescent malnutrition in South Asia, please update your aims.

Correct, and this is now clarified within the abstract. Thank you for pointing this out.

4. Please consider using one scoping review methodology and adhere to it rather than making references to 4 or 5 of them as they are not the same.

Apologies for the confusion. We now clarified the methodology that we followed and removed confusing references.

5. Please consider including examples of the proposed tables for data extraction.

Thank you for your suggestion, the supplementary tables that you have kindly suggested describe the data domains that will be extracted. However, we will make sure to publish the tables in the final review.

Best wishes

VERSION 2 – REVIEW

REVIEWER	Dr Elizabeth M. Joseph-Shehu National Open University of Nigeria, Abuja, Nigeria
REVIEW RETURNED	08-Dec-2019

GENERAL COMMENTS	The authors have revised and addressed most of the concerns raise in the paper. However, there are very few observations that still need to be addressed. - In general, the paper needs language editing. for example: - In the introduction section (Abstract), second line should be ...synthesise up to date evidence on... OR ...synthsize evidence on... - Methods and analysis section (Abstract), second line change 'The' to the - under eligibility criteria section, the concept is undernutrition. therefore, revise the first and second sentences as well as second to the last bullet in this section.
--

VERSION 2 – AUTHOR RESPONSE

Reviewer(s)' Comments to Author:

Reviewer: 2

Reviewer Name: Dr Elizabeth M. Joseph-Shehu

Institution and Country: National Open University of Nigeria, Abuja, Nigeria

Please state any competing interests or state 'None declared': None declared

Please leave your comments for the authors below

The authors have revised and addressed most of the concerns raised in the paper. However, there are very few observations that still need to be addressed.

- In general, the paper needs language editing. for example:

- In the introduction section (Abstract), second line should be ...synthesise up to date evidence on...

OR ...synthesise evidence on...

- Methods and analysis section (Abstract), second line change 'The' to the

Thank you Dr Joseph-Shehu. As you suggested, a native English speaker has reviewed now the language issues of the manuscript.

- under eligibility criteria section, the concept is undernutrition. therefore, revise the first and second sentences as well as second to the last bullet in this section.

Thank you for pointing out the consistence of the concepts.

I have made the necessary changes (highlighted in yellow).